# Tracking Topics and Frames Regarding Sustainability Transformations during the Onset of the COVID-19 Crisis

**Mariana Madruga de Brito** [1,*] , **Danny Otto** [1] **and Christian Kuhlicke** [1,2]

1 Department Urban and Environmental Sociology, Helmholtz Centre for Environmental Research, 04318 Leipzig, Germany; danny.otto@ufz.de (D.O.); christian.kuhlicke@ufz.de (C.K.)
2 Institute for Environmental Sciences and Geography, University of Potsdam, 14468 Potsdam-Golm, Germany
* Correspondence: mariana.brito@ufz.de

**Abstract:** Many researchers and politicians believe that the COVID-19 crisis may have opened a "window of opportunity" to spur sustainability transformations. Still, evidence for such a dynamic is currently lacking. Here, we propose the linkage of "big data" and "thick data" methods for monitoring debates on transformation processes by following the COVID-19 discourse on ecological sustainability in Germany. We analysed variations in the topics discussed by applying text mining techniques to a corpus with 84,500 newspaper articles published during the first COVID-19 wave. This allowed us to attain a unique and previously inaccessible "bird's eye view" of how these topics evolved. To deepen our understanding of prominent frames, a qualitative content analysis was undertaken. Furthermore, we investigated public awareness by analysing online search behaviour. The findings show an underrepresentation of sustainability topics in the German news during the early stages of the crisis. Similarly, public awareness regarding climate change was found to be reduced. Nevertheless, by examining the newspaper data in detail, we found that the pandemic is often seen as a chance for sustainability transformations—but not without a set of challenges. Our mixed-methods approach enabled us to bridge knowledge gaps between qualitative and quantitative research by "thickening" and providing context to data-driven analyses. By monitoring whether or not the current crisis is seen as a chance for sustainability transformations, we provide insights for environmental policy in times of crisis.

**Keywords:** frames; SDG; green deal; content analysis; natural language processing; NLP

## 1. Introduction

Historically, crises have motivated fundamental social, ecological, economic, and political transformations [1]. The evolving COVID-19 pandemic is no exception. In a matter of weeks, it cast a stark light on how changes can take place overnight. This includes, among others, short-term negative impacts such as an increased waste generation due to the lockdowns [2,3] but also positive environmental changes such as an improved air quality [4,5]. What remains to be understood is if fast-moving crises such as the COVID-19 pandemic can usher in lasting and far-reaching transformations [6].

Particularly at the beginning of the pandemic, an increasingly vocal debate on potential long-term changes with a rather optimistic view was initiated, with proponents arguing that the COVID-19 crisis may have opened a window of opportunity to spur on systemic ecological sustainability transformations [7]. This narrative was highlighted, for instance, by António Guterres: "We must turn the recovery from the pandemic into a real opportunity to build a better future" [8]. Similarly, multiple scholars have stated that they see an opportunity for accelerated sustainability transitions in the post-crisis period—mainly in the form of "restarting" society or by "rebuilding it better" [6,9,10]. The adherence to this narrative is echoed by both scientists and politicians. Yet, empirical evidence for such a dynamic currently lacking.

Previous experience suggests that whether or not crises result in an opportunity for initiating or accelerating sustainability transformations depends on factors such as the crisis severity, the public and media framing [11], socio-economic capacities, and political interests [7,12,13]. Hence, to understand whether or not the COVID-19 crisis can prompt sustainability transformations, it is necessary to monitor political, public, and media frames and their translation into policies. In our view, this calls for a systematic monitoring of discursive material and an in-depth study of how the relationship between the pandemic and ecological sustainability is established. This is crucial because the way environmental concerns are framed sets the stage for how they are addressed [14].

Since the beginning of the pandemic, a plethora of discursive material on COVID-19 has been produced. To varying degrees, this material has influenced the perceptions of and the reactions to the crisis. Indeed, it is widely acknowledged that traditional news media play an important role in shaping public awareness [15–17]. While social media and other forms of media have gained relevance, the analysis of newspapers is still a valuable tool for monitoring public knowledge [18]. Media coverage has been shown to influence public concern regarding climate change [19]; it can impact policy processes [20] and people's behavioural intentions [21,22]. The influence of the media in shaping opinion is even more evident in times of crisis [23]. This has been studied in relation to the Ebola outbreak in 2014 [24], the Brexit referendum [25], and the European drought of 2018 [26,27]. In this context, media news can serve as a proxy for understanding how public discourses on ongoing transformations evolve. Similarly, online search behaviour can provide insights into public awareness levels [28].

Given the multitude of discourse material available, new and variably applicable analytical approaches are needed to analyse these data, aiming to produce timely results that allow researchers to make cross-context generalisations. To this end, "big data" tools such as natural language processing (NLP) techniques [29,30] can be used to offer a "bird's eye view" of how discourses change. At the same time, context-specific and interpretative qualitative methods (i.e., "thick data" tools) are needed to elucidate the underlying patterns observed in data-driven analyses, as they allow for an in-depth analysis of prominent frames [31] (for a definition of discourse and frames see [32]).

Here, we propose the integration of NLP and qualitative content analysis on newspaper data to understand how sustainability aspects are discussed within news related to COVID-19. More specifically, we analyse how the reporting about topics underpinning sustainability has evolved in the media debate about COVID-19. The proposed approach is demonstrated by following the COVID-19 discourse on sustainability during the onset of the crisis in Germany. In a first step, we used topic modelling and quantitative content analysis to investigate the extent of the discussion on sustainability topics from March until June 2020. We also analysed how public awareness developed compared to the previous year by considering online search behaviour. In a second step, we conducted a qualitative content analysis on a smaller sample of articles, focusing on news published during the first COVID-19 wave. This allowed us to delve deeper into the discursive material and track specific frames. Emphasis was given to sustainability topics in line with the European Green Deal framework and the UN Sustainable Development Goals (SDGs) related to ecological sustainability.

## 2. Big Data and Thick Data: Methodological Considerations

Big data has been discussed as a central issue for empirical research. Some see it as a challenge for empirical social sciences [33,34]. Others weigh up the related threats and opportunities [35–38] or question whether the use of big data is truly innovative, arguing that traditional types of big data—such as administrative records and newspaper articles—have been studied for centuries [39]. Additionally, the definition of what constitutes big data has been contested, and it remains a "rather loose ontological framing" [40].

Overall, characteristics commonly associated with big data include large volumes of data that: are accessible for computational analyses, are rapidly created and shifting, have a

complex and untidy structure, and are exhaustively captured instead of sampled [35,39–41]. Due to challenges arising from pure data-driven quantitative analyses, integrating qualitative components into the research design is often advised [38,39,41,42]. In this context, the goal of mixed-methods designs can be to: (a) provide additional coverage by adding more perspectives and data on a particular issue, (b) verify results by "convergent findings" of quantitative and qualitative methods or (c) achieve "sequential contributions", meaning that what is learned in one method serves as input for the other [43]. For big data, all three aspects are applicable.

Following this rationale, we argue that the monitoring of sustainability transformations needs both big and thick data. Thick data is understood here as qualitative and detailed data which support the understanding of the context in which a pattern occurs [44]. While big data tools can help to track and describe changes in discourses over time, it is difficult to understand these transformations and the reasons behind them based solely on data-driven approaches [31]. In this regard, thick data methods can be used to add a deep-diving and context-sensitive dimension to the data-driven outcomes, enabling more complex interpretations.

Here, we propose to "thicken" the data by reducing the number of data points while enhancing the thickness of their descriptions. Hence, instead of solely providing a trend analysis of the topics mentioned in the media, we analyse how they are framed. We use the metaphor of "thick data" in reference to Geertz's "thick description" [45], which describes the need for ethnographic descriptions to go beyond recording what people are doing, as this only provides a superficial account of actual situations. Taking golfing as an example, a "thin" description portrays a person "repeatedly hitting a little round white object with a club-like device" [46]. A "thick" description sensitively interprets the context of the behaviour. It adds the frames of a golf course, gaming rules, and equipment handling in an attempt to grasp the situation fully. Thick descriptions not only enable an understanding of cultural contexts beyond "thin" statements of observable behaviour, but they also allow a further understanding of patterns emerging in big data. Conversely, big data can support sampling strategies or highlight hotspots for thick data tools. Big data can also take on the role of thick data if, for instance, qualitative text analyses are complemented by the study of large databases.

## 3. Material and Methods

### 3.1. Newspapers Sample Selection

Newspaper data were collected from a news aggregator database (wiso-net.de). The articles in our sample were published between 1 March to 30 June 2020. By using the search string "Corona*" or "Covid" or "SARS-CoV-2", 459,129 articles were retrieved. Given that the aforementioned database does not allow web scraping, the articles had to be downloaded manually. Hence, to reduce the sample size to a manageable number while at the same time ensuring topic coverage and geographical equity [26], only the newspapers with the highest circulation rates in each German federal state were considered. Hence, 2 nationwide and 19 regional newspapers were used (Table S1). This reduced the sample to 84,587 articles. The Jaccard similarity coefficient [47] was used to identify duplicate articles. A threshold of 0.7 was applied, where 0 indicates no similarity and 1 indicates full similarity. The final data set consisted of 61,514 unique records (see Figure S1 for sampling flowchart).

### 3.2. Automated Text Analysis

Before the analysis was conducted, common NLP methods were applied to clean the data. First, we removed numbers and punctuation and converted the characters to lowercase. Then, the articles' sentences were tokenised into individual words. Stop words such as articles, pronouns and prepositions were removed from the corpus (see Table S2 for additional stop words). Finally, metadata relating to the date of publication and newspaper were extracted using regular expressions. All coding was carried out in R.

### 3.2.1. Topic Modelling

In the first step, topic modelling was conducted to provide an overview of the content of the articles and investigate if sustainability issues emerged as a relevant topic. Topic modelling is an unsupervised machine-learning algorithm, where patterns of word co-occurrences are used to identify topics that describe the corpus [48]. Each topic contains a cluster of words that frequently occur together and refer to similar subjects [49].

The Latent Dirichlet allocation (LDA) was chosen to identify predominant topics in our dataset. The LDA model assumes that articles can have multiple topics. For instance, an article might be 40% about topic 1 and 60% about topic 2. The number of topics in our LDA model was defined as 70 after computing the coherence when varying the number of topics from 10 to 100. Additional stop words used are shown in Table S2. As an outcome of the LDA, a list of the most common words and the topic probabilities for each article were obtained. The topic titles were defined inductively based on the 20 most unique words from each topic (i.e., words that tend to occur mainly in a specific topic).

### 3.2.2. Quantitative Content Analysis

Pattern matching was used to classify the articles into different sustainability topics. The articles were classified according to eight environmental topics drawn from the European Green Deal framework and the UN SDGs.

The keywords (Table S3) used to classify the articles within these topics were defined based on two sustainability glossaries [50,51]. Additional terms were identified by analysing unigram frequencies and word co-occurrences (i.e., words that occur in the same sentence). Furthermore, ten experts in sustainability transformations were consulted to select unambiguous terms. Based on that, we tagged sentences where a keyword or combination of keywords occurred as related to a given topic.

Of the 61,514 unique articles, only 2343 mentioned at least one of the sustainability keywords. A random sample of 40% of the articles was read to validate the automatic classification and identify false positives and false negatives. Missclassifications found were manually corrected. In a final step, we plotted the results obtained for each of the eight topics against the German national SDG performance [52].

### 3.2.3. Bibliometric Analysis

To add a temporal perspective to the quantitative content analysis, we conducted a bibliometric analysis [53] on the whole wiso-net database, only considering German newspapers (>39 million articles, 240 newspapers). The same sustainability keywords (Table S3) were used, but with a different time frame: from August 2017 to July 2020. For these searches, the keywords related to COVID-19 were omitted. This allowed us to determine the frequencies of articles referring to sustainability over time and see if their number increased or declined during the first wave of the pandemic. The outputs were normalised according to the total number of articles per month. This reduced the bias due to the variation in the monthly number of news.

### 3.2.4. Co-Occurrence of Sustainability Topics

To analyse the topics that were reported simultaneously by the same article, a co-occurrence analysis was conducted in the sustainability dataset (n = 2343 articles). Spearman correlation coefficients were computed to assess whether they co-occur by chance or follow a pattern. Circos plots [54] were used to visualise the interdependencies between the sustainability topics.

### 3.2.5. Word Frequencies

NLP procedures, such as term-document matrices, word frequency analyses and bigram analyses, were employed. Based on that, a network analysis graph was developed to visualise how the corpus' words are associated. In addition, word clouds were created to demonstrate how the terms' usage has changed over time and according to different

sustainability topics. For the compilation of the comparison word clouds, all mentions of a word in the corpus were counted. Then, the proportional use of the word was calculated based on the total words used in all articles across the specified topic or month.

3.2.6. Public Awareness on Sustainability

In a final step, Google Trends (GT) data (trends.google.com) were used to measure public awareness on environmental topics. GT measures the search popularity in relative terms according to a random sample of all search terms used in queries within the investigated period. A detailed description of the GT measure can be found in Rousseau and Deschacht (2020). The GT database can be searched according to user-defined "search terms" and "topics". The user-defined keywords used here are shown in Table S3. Furthermore, the pre-defined "Environment" topic was considered.

To identify the effect of the COVID-19 crisis on the population's awareness regarding sustainability topics, we compared the GT search popularity indicator from March to June 2020 with data from the same period in 2019. The nonparametric Wilcoxon test was used to identify significant differences.

*3.3. Qualitative Content Analysis*

A qualitative content analysis was conducted to provide context and an in-depth analysis of the frames which connect COVID-19 and sustainability. Based on the previous steps, 461 articles were selected as a thick data sample. These articles discussed two or more of the sustainability topics considered, and hence, were more likely to add to a "window of opportunity" narrative for sustainability transformations.

Following Schreier [55] and Mayring [56], we applied a multi-step approach using a combination of inductive and deductive reasoning. First, we developed three broad categories: (a) COVID-19 impacts with implications for the environmental SDGs and Green Deal goals; (b) COVID-19 recovery measures and their relation to the environmental SDGs and Green Deal goals; and (c) COVID-19 as a window of opportunity for sustainability transformations. Two coders worked through all the articles to identify segments related to the above-mentioned categories. Disagreements were discussed in this process and code descriptions were refined.

After the first coding round and based on the discussion between the coders, sub-categories were added in vivo to create a final version of the coding scheme. In the end, eight prominent frames were identified. Once all the articles were analysed, we calculated the number of articles in each category. We also compared the frames with the automated text analysis outputs by developing a heatmap and computed the co-occurrence of frames and topics using the same method described in Section 3.2.4.

## 4. Results

*4.1. Quantitative Analysis: Media Debate on Topics Regarding Sustainability during the Early Stages of the COVID-19 Crisis*

4.1.1. Media Attention and Public Awareness on Sustainability

Overall, there was a sharp decline in the number of articles that mention sustainability-related keywords after the start of the COVID-19 pandemic in Germany (Figure 1). This could indicate that ecological concerns were brushed aside during the initial shock of COVID-19. Before the onset of the pandemic, clear peaks occurred during protests such as the Global Climate Strikes (GCS) (Figure 1).

With regard to the public awareness on sustainability, an adverse effect of the COVID-19 crisis on the online search behaviour for the topics climate change and environment was found (Figure 2). Conversely, there was an increase in searches regarding air quality, with more searches in 2020 compared to the same period in 2019.

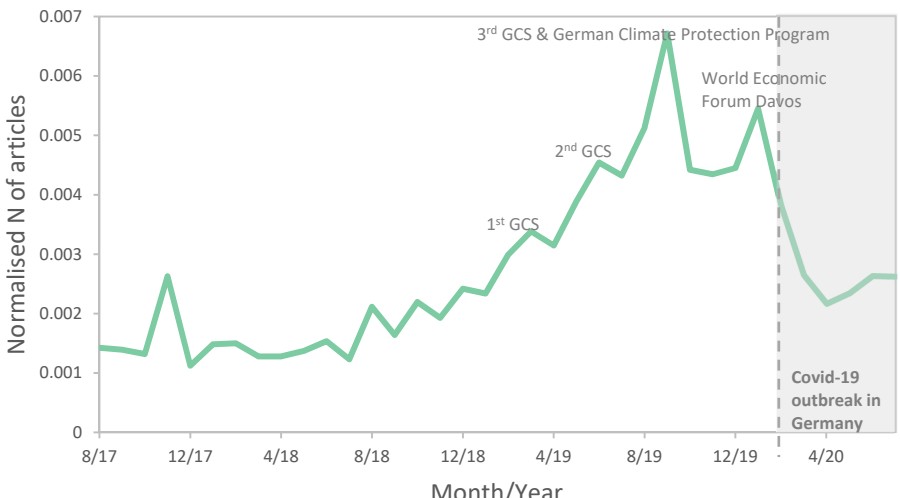

**Figure 1.** Bibliometric analysis showing the frequency of articles that mention sustainability-related keywords (Table S3) in the German media according to the wiso-net database. The data were normalised against the total N of published articles per month.

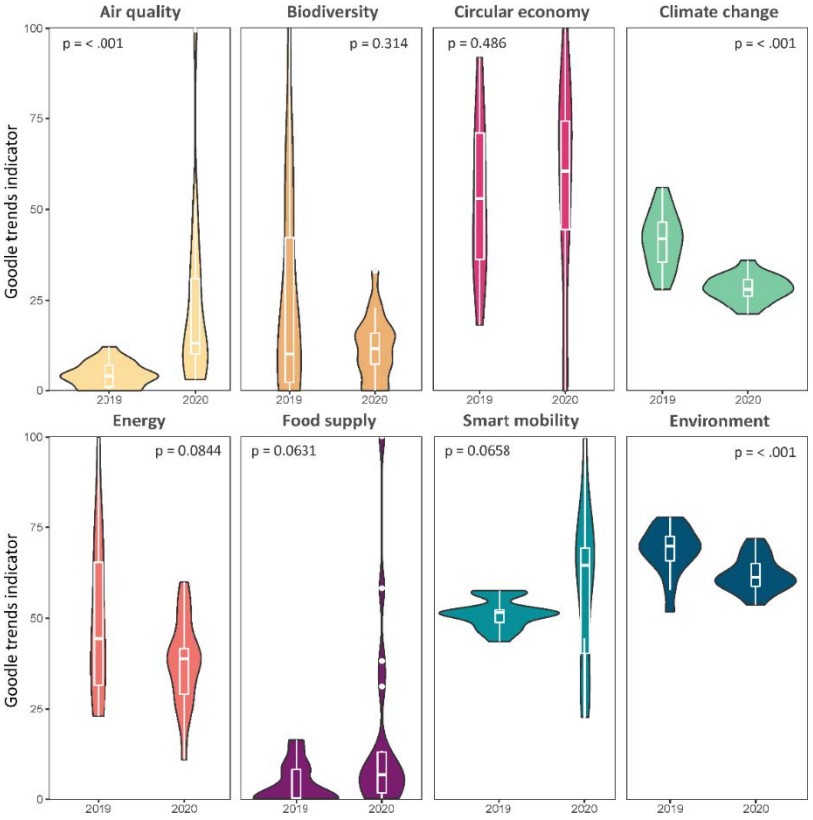

**Figure 2.** Search behaviour in 2019 and 2020 according to Google Trends data. Higher values indicate a higher amount of searches. For both years, data from March to June were considered. Wider sections represent a higher density of people searching for a term; the thinner sections represent a lower density of searches. *p*-values were obtained by using the nonparametric Wilcoxon test.

4.1.2. Media Statements about Sustainability (MSS) during the Onset of the COVID-Crisis

To analyse the relationship between sustainability and the COVID-19 crisis, only articles that mention the pandemic were considered (for sampling procedure, see Section 3.1). Topic modelling outcomes suggest that environmental problems are underrepresented in contrast to other topics (Table S4). The topics that dominate the corpus were related

mainly to family and everyday life issues (3.5%), financial insecurity (3.4%), and quarantine restrictions (3.1%). Of the 70 topics, only one is directly related to environmental issues (1.5% of the corpus, n = 932). It includes news about outdoor recreation, gardening, and smart mobility (see topic 62 in Table S4). Hence, based solely on this unsupervised machine-learning algorithm, it was not possible to detect sustainability as a core issue in the COVID-19 discourse.

To further investigate the COVID-19 discourse on sustainability, all COVID-19-related articles were classified into eight sustainability topics following the scheme presented in Table S3. Based on that, 2935 media statements about sustainability (MSS) were identified. Most of the MSS refer to climate change aspects (39%, n = 1142), followed by food supply (22%, n = 637) (Figure 3). These numbers are low compared to the number of articles included in the analysis (n = 61,514). Indeed, for each week and topic, an average of only 0.6% of the COVID-19 articles reported about sustainability issues. This suggests that the early COVID-19 discourse in Germany was not centred on sustainability issues, which was expected, because issues that directly impact people tend to become a priority in an immediate crisis.

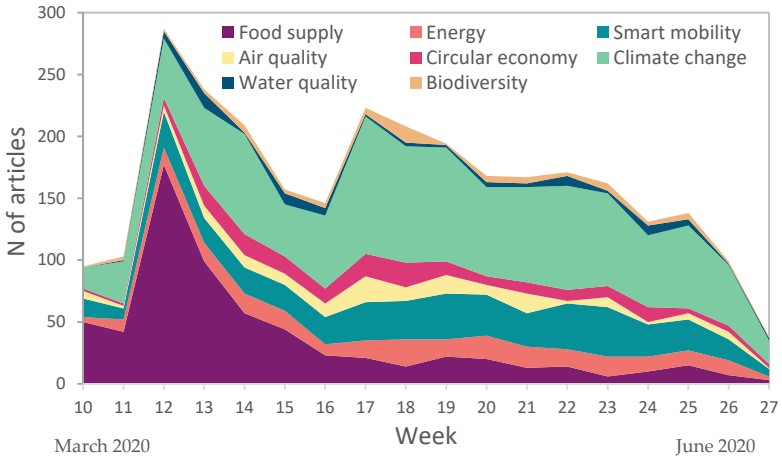

**Figure 3.** Temporal distribution of the MSS topics. The keywords used for the searches are provided in Table S3.

When considering the MSS frequency in relation to the total number of news articles, we see an increase in the proportion of climate change MSS (Figure 4). In March 2020, about 0.9% of all articles (n = 61,514) reported about climate change issues. This number increased to 2.7% in April 2020. This is also reflected in the usage of terms, which shifted drastically over time (Figure S2a). Indeed, the word stem "*Klima\**" (climate in english) appeared at a frequency of 0.0038 (words appeared 815 times across 210,311 words for this period) in April, whereas in March, the frequency was only 0.0016 (n = 411 out of 248,046 words). With regard to food supply MSS, there was higher interest in this topic in March 2020, with a rapid decline in the following months. In May, the discussion about smart mobility intensified, as reflected by the usage of the words "car", "Tesla", "railway", and "bicycles". Finally, in June, concerns regarding the "future" and the economy (e.g., "euro", "economic stimulus package") became more evident. In summary, there was a shift in the discourse, from a primary focus on food supply in March, to increasing references to a "chance" for climate change protection in April, followed by an intensified discussion on mobility in May and a focus on the restoring the economy in June.

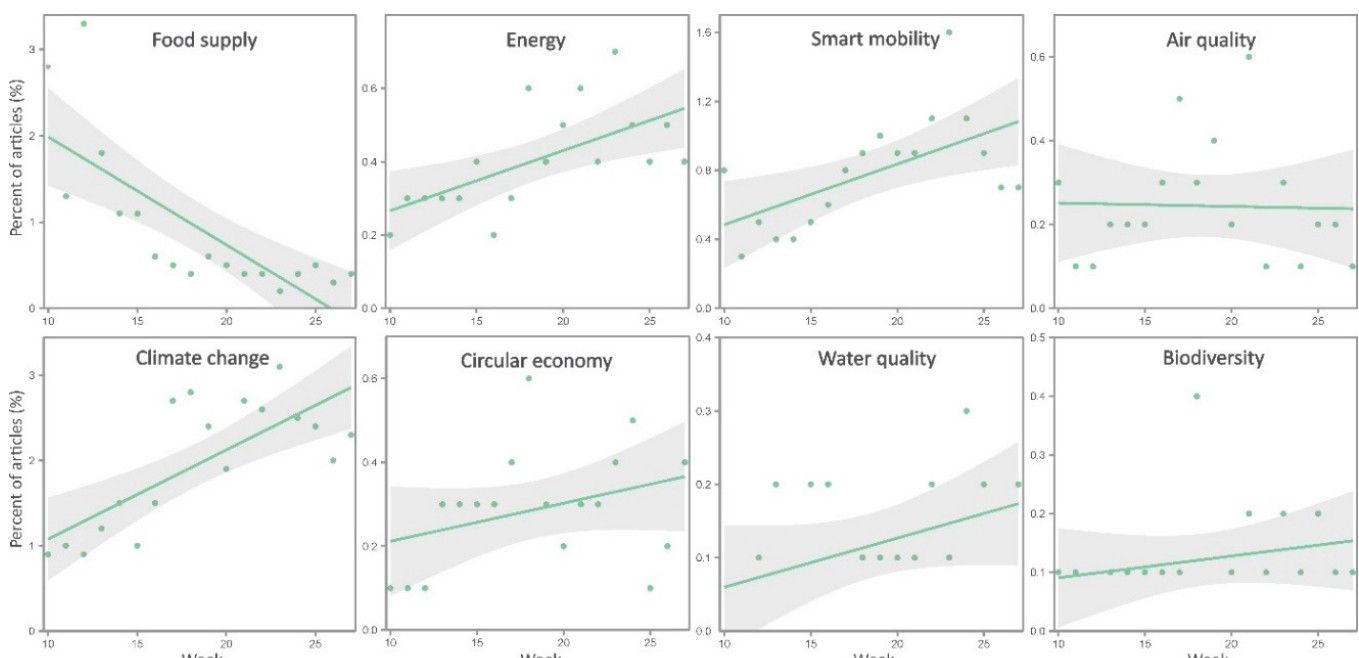

**Figure 4.** Temporal distribution of the sustainability topics with a 95% confidence interval. For each topic, the frequency of articles that contain the keywords in Table S3 in the text is shown for each week.

### 4.1.3. Co-Occurring Sustainability Topics and Keywords

To better understand the patterns behind the articles that mention multiple sustainability topics, a co-occurrence analysis was conducted. An average clustering coefficient of 0.91 was obtained, which indicates that most of the topics are connected at least in some of the articles (Figure 5). Still, the majority of the MSS were reported alone (64.1%, n = 1880). Climate change MSS were frequent drivers of network connections (present in 71.8% of all co-occurrences, n = 481). Indeed, energy, biodiversity, and air quality MSS were mentioned mainly together with climate change MSS. This is reflected in the strong correlations found between climate change and other topics (Table S5). Conversely, food supply, which was the second-most dominant MSS in our database, was primarily reported in isolation (n = 594), with few co-occurrences with climate change. This could indicate that these MSS are not directly related to sustainability but refer to individual food supply needs. This was verified when validating our classification system, where it became clear that the discourse on food supply is centred on panic buying and not on long-term food security.

To identify weakly integrated themes, a biography network was elaborated (Figure S3). It illustrates how the words in the corpus are associated. Terms related to the COVID-19 impacts cluster together (e.g., "consequences", "people", "million"). On the other hand, sustainability-related terms are side-lined (e.g., "climate change", "environment"). Several word pairs are disconnected (e.g., "Green Deal", "Fridays for Future", "renewable energy"). This indicates that there is little crossover between sustainability and the COVID-19 consequences as portrayed by the German media. It also shows that sustainability is not the central theme depicted in the corpus, even in the sample of articles that discuss at least two sustainability topics.

### 4.1.4. Results Validation

We evaluated the accuracy of the automatic classification system by conducting close reading and annotating 40% of the articles, as mentioned in Section 3.2.2. The automatic classification of the MSS was accurate in 96.1% of the cases, with a standard deviation of 2.2% (Table S6). The misclassifications correspond mostly to articles that mention sustainability-related keywords but outside of the COVID-19 context. Overall, MSS topics such as circular economy and food supply presented the highest levels of accuracy (98.9%

and 98.7%, respectively). Conversely, MSS about climate change were overestimated (accuracy of 92.3%).

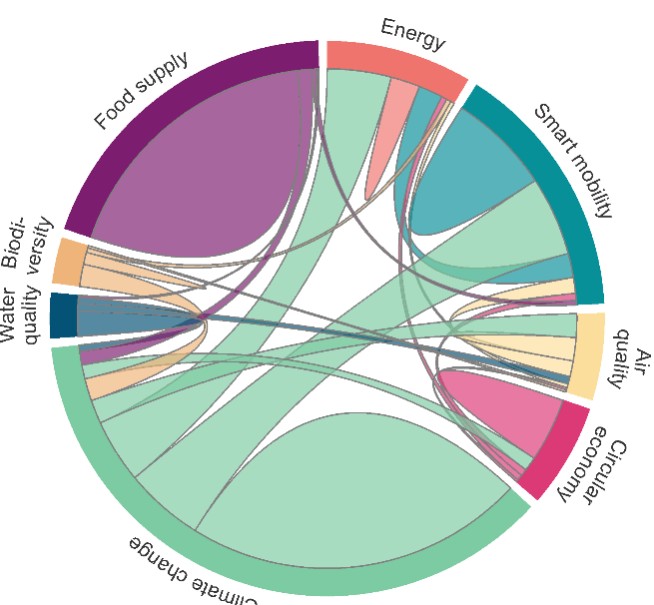

**Figure 5.** Co-occurring MSS topics. The ribbons connecting two MSS indicate copresence links, and their size is proportional to the number of links. Ribbons without connections indicate MSS that were reported alone. For clarity purposes, only co-occurrences >5 are shown. For an interactive visualisation see Supplementary Material S1.

As an additional validation measure, we built a comparison word cloud, which highlights the most unique terms in each topic (Figure S2b). The results indicate that the MSS are consistent with the sustainability topics they were assigned to. For instance, the word "water" was mentioned 168 times in the "water quality" MSS articles and only 13 times in the "biodiversity" MSS articles.

### 4.2. Qualitative Analysis: Frames Connecting COVID-19 and Sustainability

The big data bird's eye view (Figure 3) shows a decrease in coverage on sustainability since the beginning of the pandemic and only reveals a few articles that link COVID-19 and sustainability. In an in-depth interpretative content analysis, we differentiated these linkages further by analysing three bundles of frames, as shown in Table 1.

Of the 461 articles included in the qualitative analysis, 79.6% (n = 367) had substantive information related to the frames mentioned in Table 1. The following items discuss each of these frames and the main actors supporting them. Notably, these frames are interlinked and overlap, both in terms of occurrence over time (Figure S4) and content (Figure 6). Indeed, they do not occur as separate lines of discourse but instead supplement or counteract one another, suggesting that there is no dominant frame.

#### 4.2.1. The COVID-19 Crisis Led to Observable Short-Term Positive Environmental Impacts (PEIs)

About a quarter of the articles (23.4%, n = 108) highlight that the pandemic has given nature time to recover. These were connected to early scientific reports on the environmental effects of the first lockdowns. It is stated that there was an improvement in air and water quality, as well as biodiversity, due to reduced economic activities and limited human impacts on the environment. The clean canals of Venice, less nitrogen dioxide emissions in China and India, and increased sightings of wildlife are frequent points of reference for this frame. However, this frame rarely ends with this simple description. Most of the articles reference researchers and emphasise that these effects are temporary and are not enough to impact overall climate developments [7]. They might be

enough to enable Germany to accomplish its 2020 greenhouse gas emission reduction goals, as some commentators note, but they do not present a solution for the climate crisis. This line of argument is accompanied by warnings of potential rebound effects. Parallels are drawn between post-pandemic scenarios and the fast return to pre-crisis emission levels that accompanied the economic recovery after the Global Financial Crisis (2008/09). This frame is closely linked to air and water quality MSS (Figure 7).

**Table 1.** Overview of sustainability frames and percent of articles (n = 367) that mention them.

| Category | COVID-19 Sustainability Frame | % |
|---|---|---|
| COVID-19 impacts with implications for the SDGs and Green Deal goals | The COVID-19 crisis led to observable short-term Positive Environmental Impacts (PEIs) | 23.4 |
| | The COVID-19 crisis spurred Behavioural Changes that have an impact on sustainable development (BC) | 24.3 |
| COVID-19 recovery measures and their relation to the SDGs and Green Deal goals | The COVID-19 recovery programme and Measures Address or should address environmental Sustainability goals (MAS) | 27.3 |
| | The COVID-19 recovery Measures are Not Doing Enough to achieve sustainability goals (MNDE) | 7.4 |
| | The COVID-19 recovery Measures Focus on the Economy First (MFEF) | 4.6 |
| COVID-19 as a window of opportunity for sustainability transformations | The COVID-19 crisis is an Opportunity for Change towards more Sustainable development (OCS) | 25.5 |
| | The COVID-19 crisis poses Challenges for Sustainability Transformations (CST) | 21.9 |
| | The COVID-19 crisis as an Analogy for the Climate Crisis (ACC) | 6.3 |

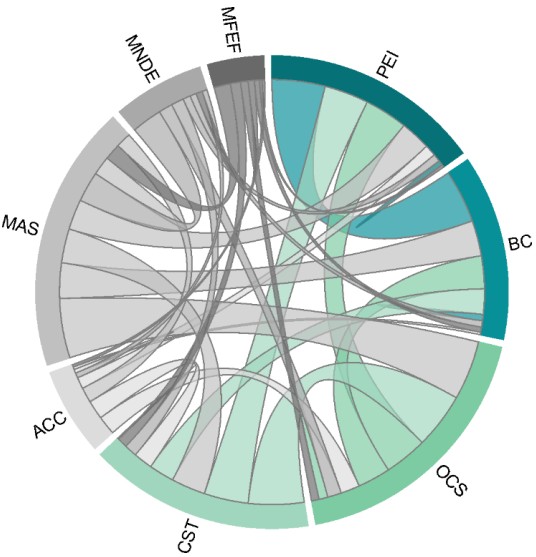

**Figure 6.** Relationships between the frames. For the acronyms, see Table 1. The ribbons connecting two frames indicate copresence of links and their size is proportional to the number of links. Ribbons without connections indicate frames that were reported alone. For an interactive visualisation, see Supplementary Material S2.

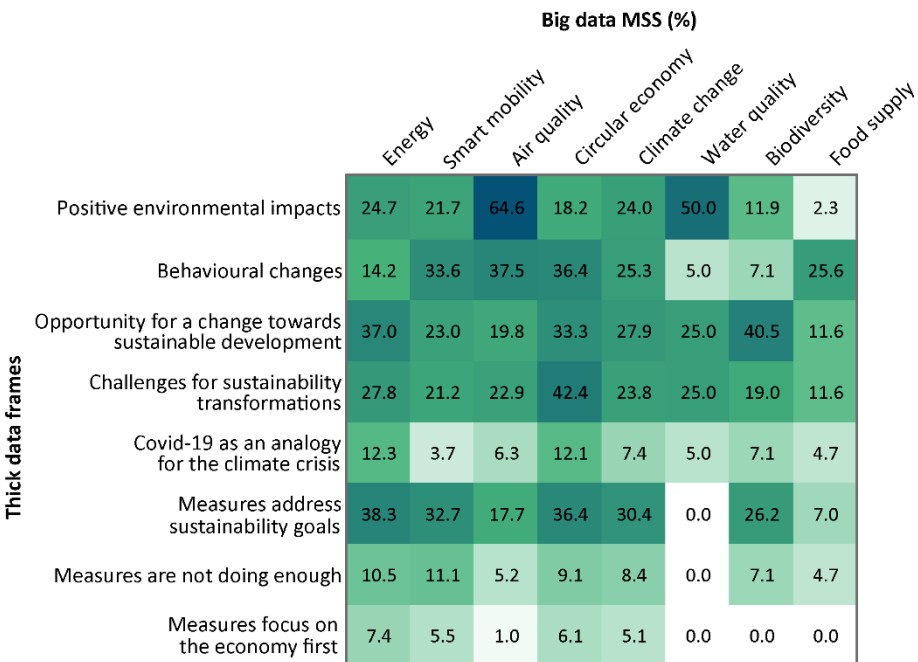

**Figure 7.** Relationships between the MSS and the frames identified in the thick data analysis. The numbers represent the percentage of articles in the subsample (n = 367) that are tagged according to an MSS and present a specific frame. For instance, 64.6% of the air quality articles in the subsample were also classified as having the positive environmental impacts (PEIs) frame.

### 4.2.2. The COVID-19 Crisis Spurred Behavioural Changes That Have an Impact on Sustainable Development (BC)

A total of 112 articles (24.3%) emphasised that the COVID-19 crisis has led to multiple behavioural changes that affect the environment either positively or negatively. These include, for instance, increased use of bicycles, mobility reduction, and a decline in oil demand. The pandemic has also affected consumption patterns (e.g., panic buying) and waste generation, as indicated by the high linkage between the BC frame and circular economy and food supply MSS (Figure 7). Similar to the positive environmental impacts (PEIs) frame, it is stated that these changes may be reversed after the crisis is over. Still, it is argued that the digitalisation of work will remain a feature of the post-COVID world, thus reducing mobility needs and overall fossil fuel consumption. Nevertheless, countervailing negative behavioural changes are also expected, such as a preference for using private cars and the avoidance of public transportation. Furthermore, experts highlight that recreational air travel is not likely to decrease after the pandemic.

### 4.2.3. The COVID-19 Recovery Programme and Measures Address or Should Address Environmental Sustainability Goals (MAS)

This is the most common frame in our sample (27.3%, n = 126). It is reasoned that the recovery plan should boost the economy while at the same time focusing on long-term strategies for advancing the climate agenda. Experts and politicians argue that the UN SDGs may act as a normative guide for selecting the targets for future investments. Overall, the ecological modernisation of transport and the car industry is a central point of this frame, as evidenced by the linkage between this frame and MSS on smart mobility (Figure 7). Of the measures discussed, the most prominent (and controversial) is the environmental bonus programme for the purchase or lease of electric or plug-in hybrid vehicles. We also found broader arguments that called for recovery programmes to focus on sustainability or include "climate-friendly" technologies and energy. From a long-term perspective, it is argued that the breakdowns of normality caused by the pandemic should be used to assess the possibilities for a more sustainable recovery (i.e., innovative, digital,

low CO2 emissions). It is noteworthy that the water quality MSS are not linked to any of the measures frames (MAS, MNDE, and MFEF) (Figure 6). This could indicate that water issues may be overlooked in the news media reporting.

### 4.2.4. The COVID-19 Recovery Measures Are Not Doing Enough to Achieve Sustainability Goals (MNDE)

This frame (7.4%, n = 34) criticises the recovery measures for not including regulations or practices that would accelerate sustainability efforts or even for hampering those efforts. Politicians and environmental NGOs question the use of environmental bonus programmes for transforming the mobility sector. Furthermore, they are critical of government rescue programmes for airline companies. They argue that investments in environmental protection projects are too low to achieve the climate goals of the Paris Agreement. This frame is usually mentioned together with the MAS frame (COVID-19 recovery programme and measures address or should address environmental sustainability goals) by pointing out existing challenges (Figure 6).

### 4.2.5. The COVID-19 Recovery Measures Focus on the Economy First—Rollback for Sustainability (MFEF)

The MFEF frame emphasises a quick economic recovery that should not be hampered by ecological concerns. This frame is consistent with policy discourses that propose halting or postponing environmental standards and goals [57,58]. Only a few newspapers (4.6%, n = 21) report on this position. Nevertheless, this number should be interpreted cautiously, as only articles that reported on at least two sustainability topics were included in the qualitative content analysis (Figure S1). This frame is supported mainly by politicians from federal states with extensive vehicle production facilities or industry sector representatives. It focuses on the situation of large industries (especially the car industry) during the pandemic and suggests rollbacks of various climate regulations (such as CO2 taxes or emission levels). The articles also mention the pressure to loosen up regulations regarding the circular economy and to postpone the adoption of environmental measures for the energy and car industries (Figure 7).

### 4.2.6. The COVID-19 Crisis Is an Opportunity for Change towards More Sustainable Development (OCS)

These articles (25.2%, n = 116) highlight that the government should consistently pursue socio-ecological transformations to deal with the aftermath of the pandemic. A central challenge mentioned is the transformation of transport, energy, and agriculture in an economically sensible, socially acceptable, and sustainable way. At the heart of the debate is the Green New Deal. Both politicians and experts emphasise that the COVID-19 crisis and the climate crisis must be tackled together. Similarly, economic institutes argue that the ongoing crisis is a good time to make longer-term investments in climate protection. Another point often mentioned is that the COVID-19 crisis has shown how society can be mobilised and how quickly changes can be made. Indeed, structures, institutions, and behaviours that were considered unchangeable were put to the test. Hence, the COVID-19 crisis is seen as an opportunity to break up ossified structures to trigger a wave of green recovery in Germany. Articles in this class also point out that the COVID-19 crisis is an opportunity for the renewable energy industry because it is accelerating the coal phase-out and, as some put it, "ending the oil age".

### 4.2.7. The COVID-19 Crisis Poses Challenges for Sustainability Transformations (CST)

Various factors that might challenge sustainability transformations are connected in this frame (21.9%, n = 101). From an economic perspective, the financial feasibility of climate objectives is questioned due to the costs of recovery programmes and the strain that COVID-19 regulations are putting on industry and the service sector. Articles that are sceptical about the successful implementation of climate protection measures or regulations in light of the pandemic tend to reference industry actors and (conservative) politicians in particular.

In addition to these economic challenges, the articles also describe a shift of attention away from climate issues. As society is focused on the immediate COVID-19 response, there are limited capacities to address the climate crisis with its uncertain timeframes. Furthermore, this loss of attention is linked to secondary effects on climate activism and (international) climate diplomacy. Focal points of this frame are the postponement of major climate conferences and changes in how climate activists stage protests (e.g., virtual formats for Fridays for Future). Both issues are seen as challenging for the sustainability efforts, as the lack of demonstrations reduces the attention given to environmental protection and the postponement of significant international meetings limits climate policy. Similar delays and shifts are reported for environmental research, technology development, and legislation processes. Interestingly, all MSS were well-represented in this frame, indicating that the challenges encompass a broad range of sustainability aspects (Figure 7).

### 4.2.8. The COVID-19 Crisis as an Analogy for the Climate Crisis (ACC)

Articles mentioning this frame (6.3%, n = 29) indicate that the COVID-19 outbreak may provide an illustrative analogy for sustainability challenges. This frame exemplifies how the COVID-19 and climate change crises require an analogous response even though they occur on different temporal scales. The analogy for the climate crisis frame is mainly supported by scientists who argue that similar to the COVID-19 pandemic, innovative and far-reaching interventions are needed to address the climate crisis [59]. These may include, for instance, the creation of new institutions and radical changes in behavioural patterns and policy governance. Along with this frame, it is stated that to curb the spread of coronavirus, ingrained societal behaviours were put to the test, including citizens´ freedom of movement. Likewise, deeply entrenched behaviours related to resource consumption and travel need to be challenged to tackle climate change. Energy and circular economy were the MSS most mentioned within this frame (Figure 7). Furthermore, even though these classes were not prominent in the MSS database (8% and 6% of the MSS, respectively), together with climate change MSS, they were the ones that were represented more strongly in the investigated frames.

## 5. Discussion

Monitoring how the media portrays different topics and frames on sustainability transformations is challenging due to the complexity of the social systems and processes involved [60] and the large amounts of discursive material available. Against this background, we proposed a mixed-methods approach [61] for following the COVID-19 discourse on sustainability in Germany. In this section, we discuss the empirical and methodological contributions of the article and the limitations of the proposed approach.

### 5.1. Empirical Findings: Evidence for a "Window of Opportunity" Narrative

Our empirical results show that the frequency of articles that mention sustainability topics in the early stages of the pandemic is limited (Figure 1). Indeed, an average of 0.6% of the articles in the corpus discussed sustainability topics each week (Figure 4). Topic modelling results indicate that (Table S4) only 1.5% of the articles were connected to environmental issues. Furthermore, google searches on sustainability on topics such as climate change, and environment were significantly lower between March and June 2020 compared to the same period in 2019 (Figure 2). This is not surprising, as during the first months of the crisis, media attention focused on the pandemic's social, economic, and epidemiological aspects rather than the climate crisis.

More broadly, these results support empirical studies that show that environmental sustainability is being overshadowed by COVID-19 [62]. Findings by [63] show that environmental issues have become less of a priority for international organisations since the COVID-19 crisis. Recent assessments indicate that the pandemic will likely undermine progress towards 12 of the 17 SDG goals [64,65]. Indeed, it is estimated that two-thirds of the 169 SDG targets are under threat due to the pandemic [66].

To discuss how our results relate to the SDGs, we linked our findings to Germany's national SDG performance [52] (Figure 8). We found that two SDGs in which Germany is not performing well were widely reported by the German news media: the SDG 13 (climate action) and the SDG 2 (food security and sustainable agriculture). A possible interpretation is that the COVID-19 crisis put the existing fragilities into the spotlight. Conversely, it could be argued that SDG 13 is intrinsically linked to all other SDGs, and the high number of SDG 2 articles is linked to panic buying during lockdowns and not necessarily connected to long-term food security. The analysis also showed that SDG 12 (sustainable consumption and production) is underrepresented in the COVID-19 media discourse, even though it is another area in which Germany is not performing well [52]. Still, the evidence indicates that COVID-19 has aggravated consumerism and the generation of waste [2,3]. This was identified in the challenge for sustainability frame (CST), where challenges for the circular economy were frequently pointed out.

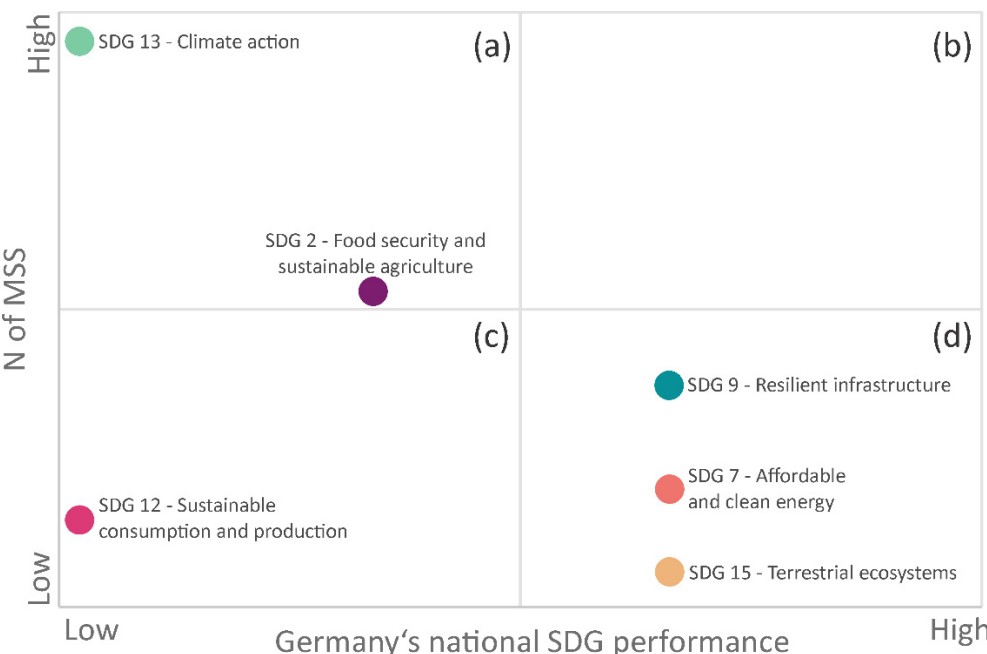

**Figure 8.** Matrix comparing Germany's national SDG performance [52] and the number of MSS. The matrix shows four patterns: (**a**) high number of MSS and low performance; (**b**) high number of MSS and high performance; (**c**) low number of MSS and low performance; and (**d**) low number of MSS and high performance.

By looking closely at the qualitative data, we found that frames supporting a more sustainable green COVID-19 recovery programme (MAS, n = 126) are more frequent than those promoting economic recovery without advocating for climate action (MFEF, n = 21). Nevertheless, this cannot be translated into unequivocal support of a "window of opportunity" frame. It became apparent in our qualitative content analysis that, even in articles that see opportunities for accelerated sustainability transformations, such acceleration does not come on its own. Both the environmental impacts (PEIs) and challenge for sustainability transformation (CST) frames point out that the pandemic could also lead to an increased environmental burden.

In sum, there certainly are thoughts about rebuilding better and initialising sustainability transformations with COVID-19 recovery measures. There is, however, no dominant frame that sees the pandemic as a chance for sustainability. In fact, the frames are closely interwoven (Figure 6). News media give contradictory representations of the relationship between COVID-19 sustainability that range from stories on environmental recreation during lockdowns to accounts that prioritise an economic recovery and see sustainability regulations as an obstacle. Depending on the actors involved, the pandemic is seen as a

chance to rebuild more sustainably, or it is perceived as an economic threat that side-lines sustainability because more urgent issues are at hand. Therefore, it is necessary to continue monitoring the discourses as the pandemic unfolds since policy priorities and opinions are expected to change based on the severity of further waves of infection.

### 5.2. Methodological Considerations: Thickening Big Data

By linking big data and thick data tools, we were able to deal with large volumes of text data; a volume that surpasses researchers' ability to conduct a close reading [67], yet the analysis remains true to the principles of traditional qualitative approaches. The use of NLP tools allowed us to scrutinise huge collections of documents across 21 news outlets and identify thematic clusters as a means of following discursive changes. Therefore, our results can be more effectively generalised. To delve deeper into the data, qualitative content analysis was employed to track specific frames.

The use of this approach allowed us to bridge knowledge gaps between qualitative and quantitative research by providing context to the "big data" through rigorous empirical qualitative research. This offers both a broad overview on relevant topics and a close analysis of particular frames. As such, the approach can yield new insights not easily achievable through traditional text mining or qualitative social science methods alone. On the one hand, NLP techniques can provide an overview of the topics that are being discussed in large-scale textual corpora through time. On the other hand, qualitative tools can yield a background to these findings by adding context. Therefore, here, we do not consider data-driven quantitative tools as an "end in itself", nor do we see traditional qualitative tools as a mere "supplement". Instead, we promote the use of both approaches alongside each other.

### 5.3. Limitations and Further Research

Despite its advancements, the approach is not without limitations. First, we relied on a stratified sample of articles as the articles need to be downloaded manually, given that the database used does not allow for web scraping (see Section 3.1). Hence, less popular newspapers were ignored. Similarly, the thick data analysis focused on articles that mentioned at least two sustainability topics (Table S3, Figure S1, n = 461). As a result, they may be skewed towards the "window of opportunity" frame.

A further drawback is that we only considered the first wave of the COVID-19 pandemic, which limits the extrapolation of the findings. With this in mind, we recommend that future studies consider how the frames on sustainability have changed during different COVID-19 waves. This will allow one to assess whether or not the pandemic has had a discernible lasting impact on the media frames used to discuss sustainability.

Future work should more deeply examine how the media debate around sustainability within the COVID-19 crisis is shaped by both the actors involved and the power relationships of the organisations that promote these discourses. Additionally, the results of the qualitative analysis can also be retrofitted to the big data analysis, to capture the extent to which the identified frames and actors were mentioned in the broad sample.

## 6. Conclusions

In this study, we examined the COVID-19-related discourse regarding sustainability transformations during the early stages of the crisis in Germany. The results show that sustainability topics were not prominent in contrast to other topics. When we looked deeper into the data, we found that there is no dominant "window of opportunity" frame that sees the pandemic as an unequivocal chance for sustainability. Some frames support a green COVID-19 recovery programme, and they appear more often than articles that support economic recovery without advocating for climate action. Still, they only represent a small fraction of the published articles. Furthermore, even in articles that perceive an opportunity for accelerated sustainability transformations, such acceleration is not taken

for granted. In fact, the presence of a window of opportunity does not necessarily imply that this opportunity will actually be taken [7].

Pragmatically, these findings are significant because reduced coverage about sustainability could dampen political will to act on the climate crisis. Indeed, as stated by Weber and Stern [17], the news media can influence people's thoughts and actions regardless of whether they are accurate or not. Even if the articles do not influence the public's opinions, the attention they bring to the topic can still be relevant as they can influence future news coverage. Thus, monitoring the immediate and long-term impacts of different COVID-19 waves on the sustainability discourse and their translation into policies remains a task for future investigation.

The mixed-methods approach described here can assist with this task by equipping scientists with a reliable toolkit for investigating discourses on rapidly evolving transformation processes. Researchers can use it to explore ongoing media discourses on sustainability transformations and detect trends in large-scale textual corpora.

**Supplementary Materials:** The following are available online at https://www.mdpi.com/article/10.3390/su131911095/s1, Table S1. Selected newspapers, number of articles per month, duplicates and articles included in the analysis, Table S2. Additional stop words for the word cloud, network and topic modelling, Table S3. Sustainability transformation topics, their related SDGs and Green Deal goals, and keywords used for coding the newspaper articles and for the google trends search, Table S4. Topics obtained by applying the LDA model to the 61,514 COVID-19-related articles. Each topic consists of a series of keywords, a topic name (assigned by the authors based on manual inspection of keywords and articles), the topic coherence and prevalence. Following LDA, one article can be assigned to more than one topic, Table S5. Correlation matrix showing Pearson correlations between the MSS, where 0 indicates a low correlation and 1 a strong correlation, Table S6. Automatic classification accuracy (%) for different MSS topics. Of the 2343 articles, 40% were randomly selected and read to determine the accuracy of the classification system, Figure S1. Sampling flowchart, Figure S2. Comparison word cloud of the changing discourse in the news media in Germany. The size of the word is proportional to its frequency. It compares the predominance of specific terms for each (a) month and (b) sustainability topic. For instance, the word "Klimawandel" (climate change) was used 158 times in April 2020 (frequency = 0.0007) and 91 times in May 2020 (frequency = 0.0004). Because this word was used more frequently in April, it only appears in the April section of the comparison cloud, Figure S3. Network analysis graph with word relationships in the MSS corpus. It portrays words that are strongly vs. weakly integrated. Each node represents a word, which must have at least 30 occurrences in the entire corpus to be present in the graph. The size and colour of the node represents the word's frequency. The darkness of the connecting lines depicts the strength of the word pair's relationship, Figure S4. Distribution of articles (n = 367) according to different frames over time. The size of the bubbles varies according to the number of articles per week.

**Author Contributions:** M.M.d.B.: Conceptualisation, methodology, software, validation, formal analysis, visualisation, data curation, writing—original draft, writing—review and editing. D.O.: Conceptualisation, methodology, validation, formal analysis, writing—original draft, writing—review and editing. C.K.: Conceptualisation, supervision, writing—review and editing. All authors have read and agreed to the published version of the manuscript.

**Funding:** This research received no external funding.

**Institutional Review Board Statement:** Not applicable.

**Data Availability Statement:** Data used for the analyses are available upon request.

**Acknowledgments:** Christian Kahmann and Tim Golla are thanked for their valuable research assistance.

**Conflicts of Interest:** The authors declare no conflict of interest.

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
