# Peer review of "Tracking Topics and Frames Regarding Sustainability Transformations during the Onset of the COVID-19 Crisis"

_sustainability, doi:10.3390/su131911095_

Round 1
Reviewer 1 Report
Dear Authors,
I find your paper very interesting, well structured and organised and well written. But first of all it is dealing with the very important issue and applying interesting methodological approach based on a broad research material.
Author Response
We would like to thank the reviewer for the positive feedback.
Reviewer 2 Report
This is an interesting and relevant article given the current context. The authors use the scientific references in the article very well. Furthermore, the methodology is clear and well explained. Results are well explored.
Author Response

(The authors gave the same response as above.)

Reviewer 3 Report
The paper “Tracking topics and frames regarding sustainability transformations during the onset of the Covid-19 crisis” has evidently been a huge undertaking and therefore an impressive piece of work that will contribute to our understanding of how to manage large amounts of data. Linking the thin data with thick data brings added value and sets the context of the large volume of data generated, helping to bring understanding to the disruptive recent events in today’s society. The limitations are also clearly explained. However, there are a few points that I believe are needed to set the piece in context and some additional explanations necessary.
- Media is mentioned in a general sense earlier on but then it becomes clear that media is referring to traditional news sites. There is no mention of the impact of social media and its potential effect on the output and readership of the major news outlets. Readership of traditional news media has been in decline, whilst I do not know the specific situation in Germany, I would assume this is also to be the case. It is perfectly legitimate to focus on the traditional news media as shown by this paper the volume of articles is large, however, it does need setting into the context of other sources of news for the general public that have gained ground over the last decade or so.
On line 62 of the three paper relating the impact of the media are from 10 and 11 years ago (Moser, 2010; Weber and Stern, 2011) and the most recent (Pérez-González, 2020) is looking at blogs, but blogs are not taken into account in the work examined. As social media has gained ground over the last decade as a source of news and even an influence on the traditional news sites then some mention of this needs to be made and justification of why traditional news sites are still important and should still be examined.
- Mention is made that empirical evidence for the opening of a window of opportunity to address sustainability challenges are scarce. Is it perhaps too early to say that? Evidence for such a dynamic will take time to emerge and time to gather that information. It could also emerge as the sum of many parts with no one dominating feature. “Currently lacking” is a more neutral way to put that. In addition, how would you know if the articles on sustainability issues changed independently of the Covid 19 articles? Would it not be possible that people were discussing these issues more but this wouldn’t be picked up in this sweep of data?
- Acronyms reduce the readability of the piece. A table of the acronyms used in the figures would be useful but not throughout the text.
- On Lines 190-192 it is stated, “This allowed us to determine the frequencies of articles referring to sustainability over time and see if their number increased or declined after the pandemic.” – The pandemic has not finished yet, do you mean after the first wave of the pandemic?
- On Lines 231-232 it is stated, “Two coders worked independently through all the articles to identify segments related to the above-mentioned categories.”
What was the purpose of using two coders working independently? Was this to gather a range of perspectives or to find agreement? What was the level of agreement and were there any differences if so, how were differences resolved?
- I find Figure 2 difficult to follow. Do you mean the density of topics mentioned? If not what is the density you refer to?
- On lines 182-183 it says “A random sample with 40% of the articles was read to validate the automatic classification and identify false positives and false negatives.” What were the results of this random sampling, how were the errors addressed?
- On Line 333 it says, “We evaluated the accuracy of the automatic classification system by reading the articles; computing the frequency of the keywords in each MSS topic.” Which articles does this refer to? Was this a sample test or a full test of all the articles selected? A flow diagram showing each stage would be helpful with the actions taken at each stage clearly depicted (systematic mapping articles may help here).
- On line 345 it says, “The big data bird’s eye view shows a decrease in coverage on sustainability” and yet in Figure 4 most of the trends were upwards. Could you clarify which data you are referring to suggest that there is an overall decrease in coverage on sustainability.
- On lines 499-502 it says “Furthermore, public awareness on sustainability on topics such as climate change and environment were significantly lower between March and June 2020 when compared to the same period in 2019 (Fig. 2).” Public awareness may not have changed, but the priorities of the public will have changed and the focus of the public will have changed. Suggest revising the sentence.
English issues
- Line 40: “usher in” not “usher to”
- Line 153: “sentences were tokenised” not “sentences where tokenised”
- Lines 182: “A random sample of 40% of the articles” not “A random sample with 40% of the articles”
- Line 292: ‘word stem “Climate*”’ – as climate is a mass noun it is a whole word and not a word stem, presumably this is different in German, therefore I suggest you put the actual word stem used with “climate” italicised in brackets.
- Check that the past tense is used throughout the text, for example Line 548-550: ‘There is, however, no dominant “window of opportunity” frame that sees the pandemic as a chance for sustainability.’ – “there was”.
- Line 563: “On the one hand, NLP” not “At one hand, NLP”
Author Response
We thank the reviewer for the constructive comments and suggestions. Below you can find the point by point reply to all comments
Comment 1: Media is mentioned in a general sense earlier on but then it becomes clear that media is referring to traditional news sites. There is no mention of the impact of social media and its potential effect on the output and readership of the major news outlets. Readership of traditional news media has been in decline, whilst I do not know the specific situation in Germany, I would assume this is also to be the case. It is perfectly legitimate to focus on the traditional news media as shown by this paper the volume of articles is large, however, it does need setting into the context of other sources of news for the general public that have gained ground over the last decade or so.
Response: Thanks for the comment. We agree with your comment and modified the text to make it clear we are dealing with traditional news outlets.
Comment 2: On line 62 of the three paper relating the impact of the media are from 10 and 11 years ago (Moser, 2010; Weber and Stern, 2011) and the most recent (Pérez-González, 2020) is looking at blogs, but blogs are not taken into account in the work examined. As social media has gained ground over the last decade as a source of news and even an influence on the traditional news sites then some mention of this needs to be made and justification of why traditional news sites are still important and should still be examined.
Response: We agree with the observation. We added to the text that, while social media and other forms of media have gained relevance, the analysis of newspapers is still a valuable tool for monitoring public knowledge (Peterson, 2021).
Peterson, Erik (2021): Not Dead Yet: Political Learning from Newspapers in a Changing Media Landscape. In: Political Behavior, 43 (1), 339–361.
Comment 3: Mention is made that empirical evidence for the opening of a window of opportunity to address sustainability challenges are scarce. Is it perhaps too early to say that? Evidence for such a dynamic will take time to emerge and time to gather that information. It could also emerge as the sum of many parts with no one dominating feature. “Currently lacking” is a more neutral way to put that. In addition, how would you know if the articles on sustainability issues changed independently of the Covid 19 articles? Would it not be possible that people were discussing these issues more but this wouldn’t be picked up in this sweep of data?
Response: Thanks for the valuable comment. We agree that it is too early to give an answer to the question. The covid-19 crisis is still evolving. We modified the text to reflect this and wrote that information is currently lacking as suggested.
Comment 4: Acronyms reduce the readability of the piece. A table of the acronyms used in the figures would be useful but not throughout the text.
Response: This is a good point. We removed the acronyms from the text in most of the cases.
Comment 5: On Lines 190-192 it is stated, “This allowed us to determine the frequencies of articles referring to sustainability over time and see if their number increased or declined after the pandemic.” – The pandemic has not finished yet, do you mean after the first wave of the pandemic?
Response: You are correct. We mean the first wave. We changed the text accordingly.
Comment 6: On Lines 231-232 it is stated, “Two coders worked independently through all the articles to identify segments related to the above-mentioned categories.” What was the purpose of using two coders working independently? Was this to gather a range of perspectives or to find agreement? What was the level of agreement and were there any differences if so, how were differences resolved?
Response: Two coders worked through to avoid subjectivity, aiming to get a wider variety of viewpoints. We changed the text to reflect this. “Two coders worked through all the articles to identify segments related to the above-mentioned categories. Disagreements were discussed in this process and code descriptions were refined. The discussions among the coders also served as a basis for the in-vivo development of sub-categories. These were applied in a second round of coding."
Comment 7: I find Figure 2 difficult to follow. Do you mean the density of topics mentioned? If not what is the density you refer to?
Response: It represents the density of the search. We changed the legend to clarify this: Wider sections in the plots represent a higher number of people searching for a term; the thinner sections represent a lower number of people searching for a term.
Comment 8: On lines 182-183 it says “A random sample with 40% of the articles was read to validate the automatic classification and identify false positives and false negatives.” What were the results of this random sampling, how were the errors addressed?
Response: Misclassification found were corrected. The validation results are presented in session 4.1.4. Results validation
Comment 9: On Line 333 it says, “We evaluated the accuracy of the automatic classification system by reading the articles; computing the frequency of the keywords in each MSS topic.” Which articles does this refer to? Was this a sample test or a full test of all the articles selected? A flow diagram showing each stage would be helpful with the actions taken at each stage clearly depicted (systematic mapping articles may help here).
Response: Thanks for the suggestion. We opted not to add a new figure as we already have too many tables and figures in the paper. Nevertheless, we clarified the sampling strategy in the text. The text now reads “We evaluated the accuracy of the automatic classification system by conducting close-reading and annotating 40% of the articles as mentioned in Section 3.2.2”
Comment 10: On line 345 it says, “The big data bird’s eye view shows a decrease in coverage on sustainability” and yet in Figure 4 most of the trends were upwards. Could you clarify which data you are referring to suggest that there is an overall decrease in coverage on sustainability.
Response: We are referring to Fig 3, which shows a decline just after the covid-19 pandemic started
Comment 11: On lines 499-502 it says “Furthermore, public awareness on sustainability on topics such as climate change and environment were significantly lower between March and June 2020 when compared to the same period in 2019 (Fig. 2).” Public awareness may not have changed, but the priorities of the public will have changed and the focus of the public will have changed. Suggest revising the sentence.
Response: We agree with the suggestion. The text now reads: “Furthermore, google searches on sustainability on topics such as climate change and environment were significantly lower between March and June 2020”
English issues
- Line 40: “usher in” not “usher to”
- Line 153: “sentences were tokenised” not “sentences where tokenised”
- Lines 182: “A random sample of 40% of the articles” not “A random sample with 40% of the articles”
- Line 292: ‘word stem “Climate*”’ – as climate is a mass noun it is a whole word and not a word stem, presumably this is different in German, therefore I suggest you put the actual word stem used with “climate” italicised in brackets.
- Check that the past tense is used throughout the text, for example Line 548-550: ‘There is, however, no dominant “window of opportunity” frame that sees the pandemic as a chance for sustainability.’ – “there was”.
- Line 563: “On the one hand, NLP” not “At one hand, NLP”
Response: Thanks for the detailed check. We corrected all typos. Furthermore, the article was proofread by a native-speaker.